# The Association of Gross Tumor Volume and Its Radiomics Features with Brain Metastases Development in Patients with Radically Treated Stage III Non-Small Cell Lung Cancer

**DOI:** 10.3390/cancers15113010

**Published:** 2023-05-31

**Authors:** Haiyan Zeng, Fariba Tohidinezhad, Dirk K. M. De Ruysscher, Yves C. P. Willems, Juliette H. R. J. Degens, Vivian E. M. van Kampen-van den Boogaart, Cordula Pitz, Francesco Cortiula, Lloyd Brandts, Lizza E. L. Hendriks, Alberto Traverso

**Affiliations:** 1Department of Radiation Oncology (Maastro), GROW School for Oncology and Reproduction, Maastricht University Medical Centre+, 6229 ET Maastricht, The Netherlands; fariba.tohidinezhad@maastro.nl (F.T.); dirk.deruysscher@maastro.nl (D.K.M.D.R.); ycp.willems@student.maastrichtuniversity.nl (Y.C.P.W.); francesco.cortiula@maastro.nl (F.C.); or traverso.alberto@unisr.it (A.T.); 2Department of Respiratory Medicine, Zuyderland Medical Center, 6419 PC Heerlen, The Netherlands; j.degens@zuyderland.nl; 3Department of Pulmonology Diseases, VieCuri Medical Centre, 6200 MD Venlo, The Netherlands; vvankampen@viecuri.nl; 4Department of Pulmonary Diseases, Laurentius Hospital, 6043 CV Roermond, The Netherlands; c.pitz@lzr.nl; 5Department of Medical Oncology, University Hospital of Udine, 33100 Udine, Italy; 6Department of Clinical Epidemiology and Medical Technology Assessment, Maastricht University Medical Center+, 6229 HX Maastricht, The Netherlands; 7Department of Pulmonary Diseases, Maastricht, GROW School for Oncology and Reproduction, Maastricht University Medical Center+, 6202 AZ Maastricht, The Netherlands; lizza.hendriks@mumc.nl; 8School of Medicine, Vita-Salute San Raffaele University, 20132 Milan, Italy

**Keywords:** non-small cell lung cancer (NSCLC), brain metastases (BM), gross tumor volume (GTV), radiomics, thoracic radiotherapy

## Abstract

**Simple Summary:**

Around 30% of patients with stage III non-small cell lung cancer (NSCLC) receiving radical chemoradiotherapy will develop symptomatic brain metastases (BM) during the disease course. Identifying risk factors for BM can help improve the management of these patients. The aim of this current study was to identify clinical risk factors, including gross tumor volume (GTV) radiomics features, for developing BM in patients with radically treated stage III NSCLC. We found that age, NSCLC subtype, and GTV of lymph nodes (GTVn) were significant factors for BM. GTVn radiomics features provided higher predictive value than GTV of primary tumor (GTVp) and GTV. Therefore, GTVp and GTVn should be separated in clinical and research practice.

**Abstract:**

Purpose: To identify clinical risk factors, including gross tumor volume (GTV) and radiomics features, for developing brain metastases (BM) in patients with radically treated stage III non-small cell lung cancer (NSCLC). Methods: Clinical data and planning CT scans for thoracic radiotherapy were retrieved from patients with radically treated stage III NSCLC. Radiomics features were extracted from the GTV, primary lung tumor (GTVp), and involved lymph nodes (GTVn), separately. Competing risk analysis was used to develop models (clinical, radiomics, and combined model). LASSO regression was performed to select radiomics features and train models. Area under the receiver operating characteristic curves (AUC-ROC) and calibration were performed to assess the models’ performance. Results: Three-hundred-ten patients were eligible and 52 (16.8%) developed BM. Three clinical variables (age, NSCLC subtype, and GTVn) and five radiomics features from each radiomics model were significantly associated with BM. Radiomic features measuring tumor heterogeneity were the most relevant. The AUCs and calibration curves of the models showed that the GTVn radiomics model had the best performance (AUC: 0.74; 95% CI: 0.71–0.86; sensitivity: 84%; specificity: 61%; positive predictive value [PPV]: 29%; negative predictive value [NPV]: 95%; accuracy: 65%). Conclusion: Age, NSCLC subtype, and GTVn were significant risk factors for BM. GTVn radiomics features provided higher predictive value than GTVp and GTV for BM development. GTVp and GTVn should be separated in clinical and research practice.

## 1. Introduction

Up to 30% of patients with stage III non-small cell lung cancer (NSCLC) receiving radical chemoradiotherapy will develop symptomatic brain metastases (BM) during the course of their disease [1]. Identifying risk factors for BM can help improve the management of these patients. Known risk factors are female sex, adenocarcinoma or non-squamous cell carcinoma histology, and advanced tumor stage [2,3,4]. Whether a higher gross tumor volume (GTV) is a risk factor for subsequent BM development remains unclear. It has been reported that GTV is a prognostic factor for locoregional control [5], progression-free survival (PFS) [6], and OS [6,7,8,9] in patients with NSCLC. Ji et al. found that GTV is not a significant risk factor for BM in patients with NSCLC (*n* = 335, *p* = 0.687) [10], whereas a larger GTV was associated with an increased risk of BM in patients with small-cell lung cancer (SCLC) (HR = 1.37, 95% CI 1.09–1.73, *p* = 0.007) [11].

Radiomics is increasingly being used in cancer risk prediction [12,13] and is, based on the seed-and-soil hypothesis, also of interest to evaluate in stage III NSCLC (primary tumor and/or involved lymph nodes) and in the BM prediction setting [14]. It is a quantitative imaging analysis technique that provides image-derived metrics capable of quantifying textures at a higher granularity, beyond the ability of the naked eye. Four studies showed that radiomic models based on pretreatment computed tomography (CT) might predict BM in NSCLC, but the models did not always add value to clinical models [15,16,17,18,19]. Important limitations of these studies were small sample sizes, inadequate baseline staging, only evaluating the primary tumor and not the involved lymph nodes, and the use of heterogeneous CT protocols. Thus, it is difficult to draw firm conclusions.

According to the backgrounds and rationales above, we conducted the current study to explore risk factors and develop prognostic models for BM in radically treated stage III NSCLC. We used a large dataset with adequately staged patients (baseline ^18^F-labeled fluorodeoxyglucose positron emission tomography–CT scan [^18^FDG-PET-CT] and brain magnetic resonance imaging [MRI]) and evaluated clinical data together with radiomics features of GTVs on the uniformly scanned planning contrast-enhanced chest CT for thoracic radiotherapy.

## 2. Patients and Methods

Patients with stage III NSCLC were retrospectively screened in five hospitals in the Netherlands and Italy (MUMC, Zuyderland, Venlo, Roermond, Udine) from 1 March 2012 to 31 July 2021. AJCC 7th edition was used for staging [20]. Eligibility criteria for this study included pathologically confirmed NSCLC, ^18^FDG-PET-CT and brain MRI performed at baseline (before antitumor therapy), treatment with chemoradiotherapy with radical intent (concurrent or sequential chemoradiotherapy, CCRT/SCRT). Exclusion criteria were participation in interventional clinical trials (NVALT-11 trial [because of prophylactic cranial irradiation [PCI] administration in one arm] [1], PET-boost trial [because of radiotherapy dose-escalation] [21], and NICOLAS trial [because of nivolumab] [22], other malignancy within 5 years before NSCLC diagnosis; surgery for NSCLC before chemoradiotherapy, and a total irradiation dose (TD) < 54Gy. Proton therapy, all types of platinum doublet chemotherapy, and adjuvant durvalumab were allowed. This study was conducted in accordance with the Helsinki Declaration of the World Medical Association and approved by the institutional review boards (W 22 01 00010). Informed consent from individuals for the use of their medical data was waived because no additional interventions were performed.

### 2.1. Acquisition of Images

The original planning CTs were retrieved from the clinical workstation database. All the images were acquired with contrast enhancement and with a slice thickness of 3 mm with consistent acquisition parameters on the two scanner manufacturers (Philips or SIEMENS). The pixel spacing varied from a minimum of 0.976 mm to a maximum of 1.52 mm in the X and Y directions.

### 2.2. Delineation of Regions of Interest (ROI)/GTV

The ROIs were the original GTVs obtained from the planning CT scan. The GTVs were delineated by a team of specialists in lung cancer radiotherapy in each slice of the planning CT based on the most recent ^18^FDG-PET information using the ARIA workstation (Varian, Palo Alto, CA, USA). GTV of the primary tumor (GTVp) and lymph nodes (GTVn) were delineated separately if anatomically distinguishable. When it was difficult to distinguish the primary tumor or lymph nodes, the tumor was contoured as either GTVp or GTVn (the choice was left to experienced radiation oncologist). GTV was calculated as the morphological union of GTVp and GTVn. The lung window setting (W = 200 HU and L = −1000 HU [Hounsfield Units]) was used to contour tumors surrounded by lung tissue and the mediastinum window setting (W = 220 HU and L = −180 HU) was applied for the contouring of lymph nodes and primary tumors invading the mediastinum or chest wall [23]. The contouring of each patient was confirmed by a senior radiation oncologist (experience >10 years).

### 2.3. GTV Radiomics Features Extraction

The pipeline for radiomic feature extraction consisted of the following steps: data conversion and pre-processing, radiomic extraction configuration and feature extraction. The data conversion was performed using an in-house Python script that converts the original DICOM and RTSTRUCT images into the .nrrd format that is mineable by pyradiomics. The Python packages simpleITK v2.1 and pyplastimatch v1.9.3 were used to convert the original DICOM CT images and the contours GTV, GTVp, GTVn into .nrrd images and corresponding binary masks. The radiomic extraction configuration included the following operations: re-sampling of the original images to the same pixel spacing of [1,1,1] using B-spline interpolation, removal of outliers from the binary masks above 3σ from the distribution of intensity values for each patient, application of wavelet filtering in all the 13 directions to generate wavelet features. The following feature categories were extracted from both original and wavelet-filtered images: first-order statistical features, and texture feature matrixes (GLCM, GLSZM, GLRLM, NGTDM). Morphological features were extracted only from the original images. The fixed-bin width approach (*n* = 25) was chosen for the quantization of statistical and texture features. The details of the source code are published on a public repository (https://github.com/Maastro-CDS-Imaging-Group/GTVNSCLC; accessed on 27 March 2023). The features were normalized to Z-score, as is common practice in statistical analyses.

### 2.4. Clinical and Treatment-Related Variables

In addition to GTV, other potential factors for BM were also recorded and investigated, including age, sex, smoking history, body mass index (BMI), performance status (PS), histology type, TNM stage at diagnosis; chemoradiotherapy type (concurrent or sequential chemoradiotherapy, CCRT/SCRT), total dose of radiotherapy, type of radiotherapy (once-daily radiotherapy, ODRT; twice-daily radiotherapy, TDRT/mix [TDRT + ODRT]), and immunotherapy.

### 2.5. Statistics

Missing data of the clinical variables were imputed by multiple imputation. Then, GTV, GTVn, and GTVp were divided into three categories by interquartile range (IQR) for risk analysis. The primary endpoint was BM confirmed by cranial imaging at any time regardless of presence of neurologic symptoms (e.g., headache or vomiting). The secondary endpoints were PFS (progression of disease at the first time in any sites confirmed by imaging or death) and OS. All endpoints were analyzed as time-to-event data from the pathological diagnosis to the respective events, which were subject to censoring at the last follow-up if no events were observed. The BM was analyzed using competing risk analysis, in which death without BM was treated as a competing event. The significant clinical risk factors (including volume of GTVs, which was excluded from the radiomic features analysis) for BM were identified using the multivariate Fine-Gray model with backward stepwise elimination [24,25].

Radiomic feature selection was performed using 1000 bootstrap resamples [26,27,28]. Within each of the 1000 bootstrap resamples, Spearman’s correlation was performed to identify and eliminate the highly correlated features (|r| >0.9). Then, the least absolute shrinkage and selection operator (LASSO) embedded with the Fine-Gray model was used to select features (lambda = 0.01). The features were sorted according to how frequently they were retained by LASSO in 1000 bootstrap resamples. We arbitrarily selected the top 13 features as the input for the backward stepwise competing risk model on the same 1000 bootstrap resamples. Then, we arbitrarily selected the top signature (more than one radiomic feature) to build the radiomic models. The coefficients were fitted using the original sample. A maximum number of five predictors was considered for each model to reduce the risk of overfitting. To build the combined model, one feature with the highest effect estimate (according to subdistribution hazard ratio [sHR]) from each model (Clinic + GTV + GTVp + GTVn) was selected.

The performance of competing-risk models at the 24 months’ time point was evaluated by area under the receiver operating characteristic curves (AUC-ROC) and calibration. The sensitivity, specificity, negative predictive value (NPV), positive predictive value (PPV), and accuracy of each model were also reported. The net-benefit decision-curve analysis (DCA) was performed to compare the models’ utility/application value [26]. A nomogram would be developed for the clinical model. If the radiomics model or combined model performed better, a nomogram would also be developed for the best one [28] (Figure 1). The time point of 24 months was chosen because most of BM develop within 2 years [1]. The effect of significant BM risk factors (features) on PFS and OS were investigated by Cox proportional hazards regression models. All tests were 2-sided, and *p* < 0.05 were considered as statistically significant. Statistical analyses were performed using R version 4.2.2 (R Project for Statistical Computing).

This type 2A study was conducted according to the Transparent Reporting of a multivariable prediction model for Individual Prognosis Or Diagnosis (TRIPOD) guideline [29]. The TRIPOD checklist was reported in Appendix A
Table A1.

## 3. Results

In total, 310 out of 524 patients were eligible, 282 of the 310 patients had available DICOM images for radiomic analysis (260 had GTVp, 254 had GTVn, 231 had GTVp + GTVn) (CONSORT diagram in Figure 2). Twenty-one patients had indistinguishable GTVp/GTVn, of which 12 were contoured as GTVn, nine as GTVp. Among the 310 patients, 54.5% were male, 51.6% had stage IIIA, and 37.4% had squamous-cell carcinoma (SCC), the median GTV was 71.2 (IQR: 35.2–115.2) cm^3^, the median GTVn was 16.4 (IQR: 0.88–244.1) cm^3^ and the median GTVp was 41.4 (IQR: 9.8–89) cm^3^ (Table 1). The median follow-up was 51.3 months (95% CI: 42.9–59.7 months), during which 176 (56.8%) patients died, 183 (59.0%) progressed, and 52 (16.8%) developed BM. The median OS was 34.8 months (95% CI: 28.4–41.2 months) and the median PFS was 19.3 months (95% CI: 15.1–23.5 months). For the 52 patients who developed BM, the median time to BM diagnosis was 10.5 months (95% CI: 9.2–11.9 months), the BM incidence at 2 years was 14.5%.

### 3.1. BM Risk Models

The clinical model identified three significant factors associated with BM: a higher age (>60 years) was protective (sHR 0.56, 95% CI 0.32- 0.99, *p* = 0.05), while non-squamous histology (sHR 2.64, 95% CI 1.28–5.46, *p* = 0.009), and a larger GTVn (median IQR: sHR 3.76, 95% CI 1.33–10.61, *p* = 0.012; upper IQR: sHR 3.86, 95% CI 1.28–11.65, *p* = 0.017) were associated with an increased risk. GTV, GTVp, the use of adjuvant durvalumab, and other clinical variables were not significantly associated with BM development (Table 2).

We extracted 861 radiomic features in total and identified five GTV features, five GTVn features, and five GTVp features that were associated with BM (Table 2). The combined model of the first top feature from each model showed that the GTVn radiomics feature (LLH glrlm Run Variance: sHR 1.53, 95% CI 1.05–2.24, *p* = 0.028) and the GTVp radiomics feature (HLH glszm Grey Level Non Uniformity: sHR 1.52, 95% CI 1.29–1.79, *p* < 0.001) were significantly associated with an increased risk of BM development, while the clinical variable (volume of GTVn) and GTV radiomics feature were not (Table 2).

### 3.2. Evaluation of the Models’ Performance

The AUCs at each time point showed that the GTVn radiomics model performed best to discriminate the patients that did and did not develop BM (AUC range: 0.71–0.74) (Figure 3A).

At 24 months, the GTVn radiomics model had the highest AUC (0.74, 95% CI: 0.71–0.86), sensitivity (84%), and the NPV (95%); the GTVp radiomics model has the highest specificity (80%), PPV (34%), and accuracy (76%) (Table 3). The calibration plot also visually showed that the GTVn radiomic model had the best calibration within 24 months (Figure 3B). The decision curve analysis showed that compared with the other models, the GTVn radiomic model provided a better net benefit for the threshold probabilities smaller than 0.3 (Figure 3C). Therefore, a nomogram was developed for the clinical model (Figure 3D) and the GTVn radiomics model (Figure 3E), respectively.

### 3.3. OS, PFS

The above factors and radiomics features were checked for their impact on PFS and OS. A larger GTVn was significantly associated with decreased OS (median IQR: HR 1.49, 95% CI 1.01–2.21, *p* = 0.045; upper IQR: HR 2.32, 95% CI 1.50–3.59, *p* < 0.001) and PFS (median IQR: HR 1.93, 95% CI 1.30–2.85, *p* = 0.001; upper IQR: HR 2.08, 95% CI 1.31–3.30, *p* = 0.002). Patients with non-squamous carcinoma were at higher risk for progression (HR 1.35, 95% CI 1.00–1.83, *p* = 0.05), but no significant association with OS was found. Age was not significantly correlated with OS or PFS (Table A1 and Table A2).

All the five GTV radiomics features were not correlated with OS. HLH first-order Median of the GTV was correlated with PFS (HR 0.87, 95% CI 0.78–0.98, *p* = 0.02) (Table A1 and Table A2).

LLH glrlm Run Variance (HR 1.20, 95% CI 1.04–1.39, *p* = 0.016) and HLH glszm Small Area Low Grey Level Emphasis of the GTVn (HR 0.78, 95% CI 0.63–0.97, *p* = 0.026) was associated with OS, but not PFS (Table A2 and Table A3).

LLL glcm Imc1 of GTVp was correlated with OS (HR 1.32, 95% CI 1.09–1.60, *p* = 0.005) and PFS (HR 1.38, 95% CI 1.15–1.65, *p* = 0.001). HLH glszm Grey Level Non Uniformity of GTVp was correlated with OS (HR 1.17, 95% CI 1.00–1.37, *p* = 0.047) but not PFS. LLH glcm Cluster Shade of GTVp was correlated with PFS (HR 1.16, 95% CI 1.02–1.33, *p* = 0.026) but not OS. (Table A2 and Table A3).

The combined model showed that HLH glszm Grey Level Non Uniformity of GTVp was correlated with OS (HR 1.17, 95% CI 1.02–1.33, *p* = 0.023) but not PFS. GTVn was associated with PFS (median IQR: HR 1.63, 95% CI 0.99–2.69, *p* = 0.054; upper IQR: HR 2.37, 95% CI 1.28–4.38, *p* = 0.006) but not OS (Table A2 and Table A3).

## 4. Discussion

Although it has been reported that the GTV volume is associated with OS and PFS, the association of the GTV volume and the subsequent risk of BM development is unclear in patients with radically treated stage III NSCLC. Our study indicated that a larger GTVn was a risk factor for BM, OS, and PFS in patients with stage III NSCLC, but GTV and GTVp were not significantly associated with BM. In contrast, Ji et al. reported that GTV was not a significant risk factor for BM (*p* = 0.687) [10]. A possible explanation could be that they analyzed the BM risk factors using Cox regression, which does not consider the competing event of death. Additionally, GTVp and GTVn were not specified. To the best of our knowledge, this is the first work reporting that GTVn is associated with subsequent BM development, while GTVp and GTV are not. The finding that lymph nodes involvement is more prognostic than the primary tumor volume warrants validation in further studies. A biological explanation could be that lung cancer cells are already more aggressive when migrating to lymph nodes and that the volume of GTVn correlates with the aggressiveness.

Interestingly, colleagues in Denmark investigated GTVp and GTVn for the first failure site in patients with locally advanced NSCLC. They found that neither GTVp nor GTVn was significantly correlated with first failure site (either locoregional failure or distant metastases) [30]. In this study, the main idea of separating GTVn from GTVp was similar to ours. However, patients with stage I-II or stage IV (20.5%) were also included, and they only explored the associations with the first failure site without specifying BM or metastases to other organs. We focused on BM, regardless of whether the brain was the first site of failure and we also evaluated the association with PFS and OS. This approach is more inclusive and therefore of greater clinical practice value, as patients can still develop BM after extracranial progression.

In line with other studies [2,3,4,31], we also found that higher age (*p* = 0.05) and squamous cell carcinoma (*p* = 0.009) were independent protective factors for developing BM, while smoking history, thoracic radiotherapy dose, and the use of adjuvant durvalumab were not significantly associated with the development of BM (*p* > 0.05). The latter was in contrast to the PACIFIC trial, in which the percentage of patients with BM halved in the durvalumab arm compared with placebo (31/476 [6.5%] vs. 28/237 [11.8%], *p* = 0.015) [32]. A possible explanation could be that in the PACIFIC study only fit patients without disease progression after CCRT were selected [33], while we included NSCLC patients who had stage III at the initial diagnosis, patients who progressed after CCRT were not excluded. In addition, in the PACIFIC trial, brain MRI was not required (brain CT was allowed) and PET-CT was not mandatary, while in this current study, only patients who underwent baseline PET-CT and brain MRI to fully stage and exclude occult BM were included.

As far as we are aware, this is the first study which found that the GTVn radiomics model performed better than the clinical, GTVp radiomics, GTV radiomics, and combined model, with the highest AUC (0.74), sensitivity (84%), and NPV (95%). Consistent with earlier studies, the GTVp radiomics model was not as good as the clinical model [18]. Generally, a high sensitivity leads to a better ability of ruling a disease out, and a high specificity leads to a better ability of ruling in a disease [34]. As our aim of this study is to identify patients who are at higher risk of developing BM, it is better in ruling out BM (higher sensitivity) than ruling in BM, which indicates that management to prevent or detect BM (such as prophylactic cranial irradiation [PCI] and regular brain MRI surveillance) is needed for these patients. Therefore, although the GTVp radiomics model had a better specificity, we considered the GTVn radiomics model as a better one. Furthermore, an NPV of 95 % is very good. Although the PPV and overall accuracy of the GTVn model were not great, the model may still be very valuable in clinical practice, as it can predict with a high likelihood that a patient will not develop BM and hence should not be considered for PCI (currently only within a clinical trial) nor for brain image follow-up. In addition, the specificity and sensitivity do not change when the incidence/prevalence changes, while the PPV and NPV are dependent on the prevalence/incidence of the outcome [34]. The relatively low PPV of all the models is mainly because of the relatively low incidence of BM (14.5% at 2 years) in this cohort, which was probably due to better staging (PET-CT and MRI were performed at diagnosis of stage III NSCLC). Therefore, the GTVp and GTVn should be separately delineated and analyzed in clinical practice and related studies.

Our results showed that wavelet features were the most prominent class associated with BM development as well as OS and PFS, independently from the region of interest of choice (GTV, GTVp, GTVn). Wavelet features decompose the original CT scan into a frequency space (like an “MR-like” image) and they are able to quantify granular textures based on the differences among harder and softer tissues. Our results showed that a combination of high-pass (HLH) and low-pass filtered (LLH) wavelet features are capable of quantifying tumor heterogeneity, which is a potential surrogate for a higher tumor aggressiveness. Although in the literature there is a lack of understanding about the biological meaning of radiomic features, our results suggest that tumors (and more specifically lymph nodes) that have more enhanced textures (GLSZM features) are more likely to spread to the brain, probably suggesting a higher proliferation of aggressive cells. Additionally, there are few studies that focused on extracting the radiomic features from both the GTVp and GTVn. However, previous radiomic studies [35,36] have shown that for the prediction of distant metastases it is better to focus on a larger region than just the GTVp, the so-called peri-tumoral ring.

In addition, we provided nomograms for the clinical model and the GTVn radiomics model, together with the codes for extracting radiomics features from GTVs on the planning CT scan, which clinicians and researchers could use for conducting future studies.

Strengths of this study are the relatively large dataset, the gold standard staging (baseline PET-CT and brain MRI), the administration of radical treatment to every patient, the inclusion of immunotherapy data in a real-world setting, and the long follow-up. All GTVs were rigorously contoured and evaluated by a team of specialists in lung cancer radiotherapy. Planning CTs were homogeneous regarding the scanning protocol. One limitation lies in the fact that we used the planning CT rather than the staging CT before anti-tumor treatment. In CCRT, most patients already had received one chemotherapy administration before having the planning CT. In SCRT, only patients with a reasonable performance status and without progression after chemotherapy were sent for thoracic radiotherapy. One can question the necessity of predicting the risk of BM in patients intended to undergo SCRT but not eligible (progression, poor PS) to undergo thoracic radiotherapy. On the other hand, the use of expertly contoured GTVs is reliable as well as convenient because no additional contouring is necessary, and therefore extracting radiomics features from GTVs is feasible in clinical application for all patients with a radiotherapy treatment plan. Another limitation is the lack of external validation. To overcome this limitation, we performed bootstrapping 1000 times and LASSO regression to develop the radiomics models. Our results can be further tested in future external validation studies (evaluating our model on a separate dataset, TRIPOD type 4 studies), or further confirmed using the same methods with a larger sample size training dataset and an independent validation dataset (TRIPOD type 3 studies) [29].

## 5. Conclusions

To our knowledge, this is the first study that demonstrates the prognostic value of the GTVn volume on BM development in patients with stage III NSCLC. Younger patients and those with non-squamous cell carcinoma are at higher risk of developing BM. Radiomics features of GTVn have greater prognostic value than GTVp and GTV for BM development. Therefore, the GTVp and GTVn will be contoured and analyzed separately in clinical practice and future studies.

## Figures and Tables

**Figure 1 cancers-15-03010-f001:**
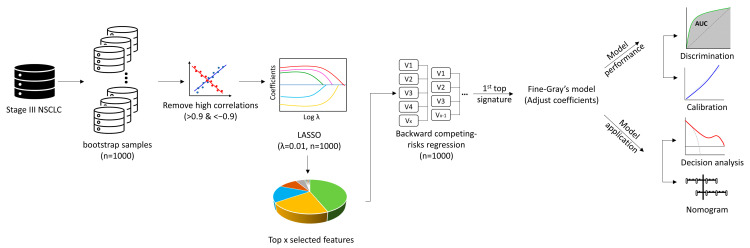
Analysis pipeline of radiomics models. This figure shows the analysis pipeline for development and evaluation of the radiomics prediction models: I, resample by 1000 bootstrap; II, eliminate highly correlated features; III, select features by LASSO regression embedded with the Fine−Gray model; IV, select top features retained by LASSO in 1000 bootstrap resamples; V, evaluate the features’ associations with BM using backward stepwise competing risk model; VI, select the top signatures to build the radiomic models using the original sample; VII, evaluate the performance of the models by AUC and calibration curve; VIII, evaluate the utility of the models by the net−benefit decision−curve analysis; IX, develop a nomogram for the best model. *Abbreviations:* AUC, area under the receiver operating characteristic curves; LASSO, least absolute shrinkage and selection operator; NSCLC, non-small cell lung cancer.

**Figure 2 cancers-15-03010-f002:**
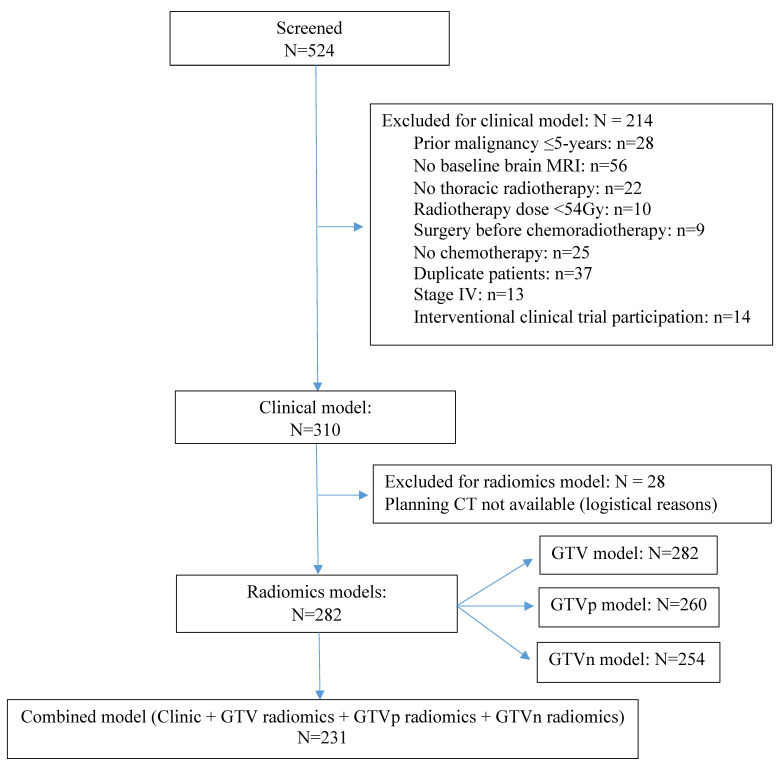
CONSORT diagram.This diagram shows the patients screening and available sample size for each model. *Abbreviations:* CT, computed tomography; GTV, gross tumor volume; GTVp, GTV of the primary tumor; GTVn, GTV of the involved lymph nodes; MRI, magnetic resonance imaging.

**Figure 3 cancers-15-03010-f003:**
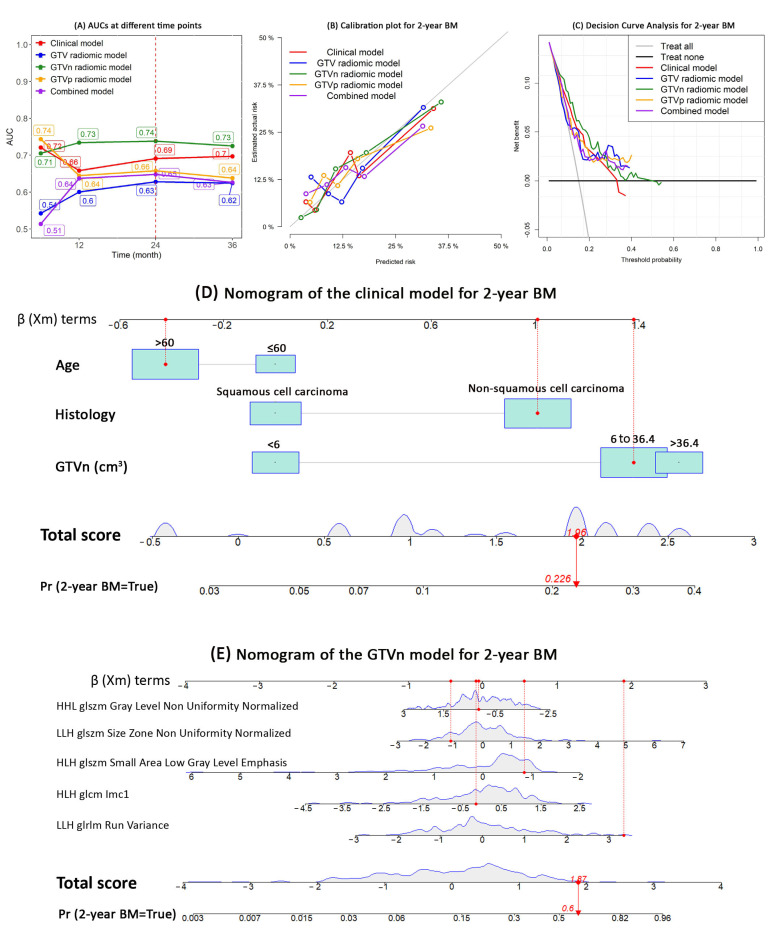
Models performance and nomogram. This figure shows the performance of the competing risk models for BM development in patients with radically treated stage III NSCLC (clinical, GTV, GTVn, GTVp, and combined models): (**A**) AUC; (**B**) Calibration plots; (**C**) Net—benefit decision curves; (**D**) nomogram of the clinical model; (**E**) nomogram of the GTVn radiomics model. *Abbreviations:* AUC, area under the receiver operating characteristic curves; BM, brain metastases; GTV, gross tumor volume; GTVp, GTV of the primary tumor; GTVn, GTV of the involved lymph nodes; NSCLC, non—small cell lung cancer.

**Table 1 cancers-15-03010-t001:** Patients characteristics (*n* = 310).

Characteristics	Number (%)
Age	
Mean ± SD	65.7 ± 8.4
≤60	80 (25.8)
>60	230 (74.2)
Male gender	169 (54.5)
Body mass index-kg/m^2^	
Normal (18.5–24.9)	137 (44.2)
Underweight (<18.5)	13 (4.2)
Overweight (25.0–29.9)	113 (36.5)
Obese (≥30)	47 (15.2)
Smoking	
Never/Former	174 (56.1)
Current	136 (43.9)
Performance status	
0	123 (39.7)
1	160 (51.6)
2–3	27 (8.7)
TNM_T	
0/X/1/2/3	171 (55.2)
4	139 (44.8)
TNM_N	
0–1	38 (12.3)
2	210 (67.7)
3	62 (20.0)
Stage	
IIIA	160 (51.6)
IIIB	150 (48.4)
Histology	
Squamous-cell	116 (37.4)
Non-Squamous-cell	194 (62.6)
Chemoradiotherapy	
Concurrent	277 (89.4)
Sequential	33 (10.6)
Type of radiation	
OD	205 (66.1)
TD/TD+OD	105 (33.9)
Total dose (Gy)	
≤66	226 (72.9)
>66	84 (27.1)
Adjuvant immunotherapy	84 (27.1)
GTV (cm^3^)	
Median (range)	71.2 (4.3–1252.8)
<35.2	78 (25.2)
35.2–115.2	153 (49.4)
>115.2	79 (25.5)
GTVn (cm^3^)	
Median (range)	16.4 (0–244.1)
<6	78 (25.2)
6–36.4	154 (49.7)
>36.4	78 (25.2)
GTVp (cm^3^)	
Median (range)	41.4 (0–1195.6)
<9.8	78 (25.2)
9.8–89	154 (49.7)
>89	78 (25.2)

*Abbreviations:* GTV, gross tumor volume; GTVp, gross tumor volume-primary lung tumor; GTVn, gross tumor volume-metastatic lymph nodes; SD, standard deviation.

**Table 2 cancers-15-03010-t002:** BM competing risk models.

	sHR	95% CI	*p*
**Clinical model** (*n* = 310, 52 BM)			
Age (>60 vs. ≤60)	0.56	0.32–0.99	**0.045**
Histology (Non-Squamous vs squamous)	2.64	1.28–5.46	**0.009**
GTVn (cm^3^)			
<6	[Reference]		
6–36.4	3.76	1.33–10.61	**0.012**
>36.4	3.86	1.28–11.65	**0.017**
**GTV radiomic model** (*n* = 282, 46 BM)			
HLH firstorder Median	0.63	0.50–0.78	**<0.001**
HLH glcm Imc1	1.72	1.13–2.61	**0.011**
Original firstorder Skewness	1.39	1.17–1.64	**<0.001**
Original glszm Zone Entropy	0.66	0.50–0.87	**0.003**
HHH glszm Small Area Emphasis	1.66	1.17–2.35	**0.004**
**GTVn radiomic model** (*n* = 254, 44 BM)			
LLH glrlm Run Variance	1.77	1.26–2.48	**0.001**
HLH glcm Imc1	1.67	1.14–2.46	**0.009**
HLH glszm Small Area Low Grey Level Emphasis	0.58	0.38–0.89	**0.012**
LLH glszm Size Zone Non Uniformity Normalized	1.54	1.27–1.87	**<0.001**
HHL glszm Grey Level Non Uniformity Normalized	0.67	0.48–0.93	**0.018**
**GTVp radiomic model** (*n* = 260, 39 BM)			
LHH glszm Small Area Low Grey Level Emphasis	1.66	1.29–2.14	**<0.001**
LLH glcm Cluster Shade	1.40	1.10–1.80	**0.007**
HLH glszm Grey Level Non Uniformity	1.90	1.58–2.29	**<0.001**
HLL firstorder Root Mean Squared	0.62	0.43–0.90	**0.013**
LLL glcm Imc1	1.93	1.06–3.51	**0.032**
**Combined model** (*n* = 231, 37 BM)			
GTVn (cm^3^)			
<6	[Reference]		
6–36.4	3.09	0.68–13.98	0.143
>36.4	2.49	0.50–12.53	0.268
GTV HLH glcm Imc1	1.33	0.82–2.16	0.242
GTVn LLH glrlm Run Variance	1.53	1.05–2.24	**0.028**
GTVp HLH glszm Grey Level Non Uniformity	1.52	1.29–1.79	**<0.001**

*Abbreviations:* BM, brain metastases; CI, confidence interval; GTV, gross tumor volume; GTVp, gross tumor volume-primary lung tumor; GTVn, gross tumor volume-metastatic lymph nodes; sHR, subdistribution hazard ratio.

**Table 3 cancers-15-03010-t003:** Model Performance at 24-months.

Models	AUC (95% CI)	Sensitivity	Specificity	PPV	NPV	Accuracy
Clinical	0.69 (0.66–0.82)	59%	77%	33%	91%	74%
GTV	0.63 (0.57–0.77)	65%	67%	27%	91%	67%
GTVn	0.74 (0.71–0.86)	84%	61%	29%	95%	65%
GTVp	0.66 (0.62–0.81)	54%	80%	34%	90%	76%
Combined	0.65 (0.60–0.78)	70%	60%	25%	91%	62%

*Abbreviations:* GTV, gross tumor volume; GTVp, gross tumor volume—primary lung tumor; GTVn, gross tumor volume—metastatic lymph nodes; AUC, area under the curve; CI, confidence interval; PPV, positive predictive value; NPV, negative predictive value.

## Data Availability

Data sharing is possible upon reasonable request to the corresponding author and with the approval of institutional review board IRBs. The details of the radiomics source code are published on a public repository (https://github.com/Maastro-CDS-Imaging-Group/GTVNSCLC; accessed on 27 March 2023).

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
