# Peer review of "The Association of Gross Tumor Volume and Its Radiomics Features with Brain Metastases Development in Patients with Radically Treated Stage III Non-Small Cell Lung Cancer"

_cancers, 2023, doi:10.3390/cancers15113010_

Round 1

Reviewer 1 Report

This is a well written manuscript and timely topic given increasing interest in the use of radiomics for prognostication. Interesting findings on GTVn, lower age, and non-squamous histology being associated specifically with higher likelihood of brain metastases. Valid points are raised with respect to prior analyses not noting a correlation between GTV and BM development if not considering death as a competing event.

My only small comment is that the quality of image of Figure 1 is quite blurry and should be reformatted.

Author Response

Thank you very much for your nice comments!
We have updated the Figure. 

Reviewer 2 Report

This study investigated clinical risk factors, including gross tumor volume (GTV) radiomics features, for developing brain metastases (BM) in patients with radically treated stage III non-small cell lung cancer (NSCLC). The study found that age, NSCLC subtype, and GTVn were significant risk factors for BM. The study included 310 patients with stage III NSCLC who had been treated with radical thoracic radiotherapy.

·         Of the 310 patients, 52 (16.8%) developed BM.

·         Three clinical variables (age, NSCLC subtype, and GTVn) and five radiomics features from each radiomics model were significantly associated with BM.

·         Radiomic features measuring tumor heterogeneity were the most relevant.

·         The GTVn radiomics model had the best performance, with an AUC of 0.74.

·         The study concluded that age, NSCLC subtype, and GTVn were significant risk factors for BM. GTVn radiomics features provided higher predictive value than GTVp and GTV for BM development. GTVp and GTVn should be separated in clinical and research practice.

1.      Fig. 1 , fig. 2 and fig.3 are very blurry. They’re too hazy to see.

2.      In line with other studies [2-4, 31], we also found that higher age and squamous cell carcinoma were independent protective factors for developing BM. (line 309)
Cox models for OS and PFS were seen. Please specify the detailed results of multivariate analyses and listed the p value accordingly.

Author Response

Thank you very much for your comments!

We have updated the figures and listed the p values.